# Extracting Training Data from Molecular Pre-trained Models

**Renhong Huang**[1,2†] , **Jiarong Xu**[2*], **Zhiming Yang**[2], **Xiang Si**[2],
**Xin Jiang**[3], **Hanyang Yuan**[1], **Chunping Wang**[4], **Yang Yang**[1]
[1]Zhejiang University, [2]Fudan University, [3]Lehigh University, [4]Finvolution Group
{renh2, yuanhanyang, yangya}@zju.edu.cn,
{jiarongxu, zmyang20}@fudan.edu.cn, xsi21@m.fudan.edu.cn
xjiang@lehigh.edu, wangchunping02@xinye.com

## Abstract

Graph Neural Networks (GNNs) have significantly advanced the field of drug discovery, enhancing the speed and efficiency of molecular identification. However, training these GNNs demands vast amounts of molecular data, which has spurred the emergence of collaborative model-sharing initiatives. These initiatives facilitate the sharing of molecular pre-trained models among organizations without exposing proprietary training data. Despite the benefits, these molecular pre-trained models may still pose privacy risks. For example, malicious adversaries could perform data extraction attack to recover private training data, thereby threatening commercial secrets and collaborative trust. This work, for the first time, explores the risks of extracting private training molecular data from molecular pre-trained models. This task is nontrivial as the molecular pre-trained models are non-generative and exhibit a diversity of model architectures, which differs significantly from language and image models. To address these issues, we introduce a molecule generation approach and propose a novel, model-independent scoring function for selecting promising molecules. To efficiently reduce the search space of potential molecules, we further introduce a Molecule Extraction Policy Network for molecule extraction. Our experiments demonstrate that even with only query access to molecular pre-trained models, there is a considerable risk of extracting training data, challenging the assumption that model sharing alone provides adequate protection against data extraction attacks. Our codes are publicly available at: https://github.com/renH2/Molextract.

## 1 Introduction

Deep learning has revolutionized various scientific disciplines, inspiring researchers to adopt these advanced techniques in drug discovery to accelerate molecule identification while reducing costs. Molecules are commonly represented by molecular graphs, capturing essential structural information. Consequently, Graph Neural Networks (GNNs) have demonstrated effectiveness in tasks like property prediction [13, 59], drug discovery [53, 32], and drug design [34]. However, training these GNNs faces a significant challenge known as "data hunger" [17]; that is, a substantial amount of molecular data is required for training. For instance, developing a new drug often involves understanding intricate molecular behaviors and responses, which can only be achieved through the analysis of extensive molecular data.

---

†This work was done when the author was a visiting student at Fudan University.
*Corresponding author.

38th Conference on Neural Information Processing Systems (NeurIPS 2024).

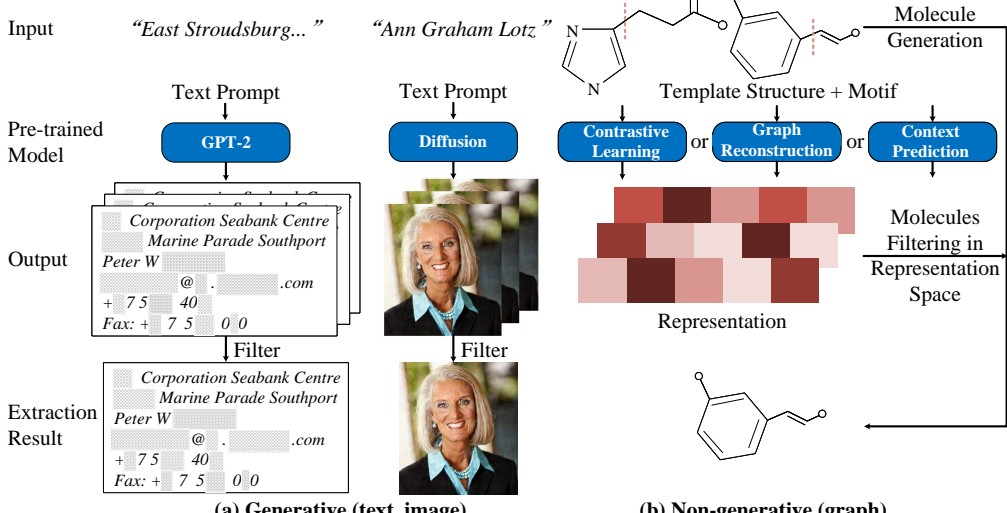

**(a) Generative (text, image)**          **(b) Non-generative (graph)**

Figure 1: Data extraction attacks across text, image, and graph. **(a)** In domains like text and image, by inputting specific text prompts, private training data can be directly extracted from the outputs generated by models. **(b)** Conversely, in the graph domain, the pre-trained models are typically non-generative, and exhibit a diversity of pre-training tasks, such as contrastive learning, graph reconstruction, and context prediction.

This massive data requirement often exceeds what any single organization can collect and maintain on its own. This limitation leads to the necessity of collaborative efforts. Nevertheless, the direct sharing of data often raises concerns about commercial confidentiality and information privacy [8, 42]. In view of this, various graph pre-training techniques emerge as viable solutions. These techniques have demonstrated remarkable generalizability across various molecular datasets [54, 19, 22, 52], facilitating *model-sharing collaboration*. Organizations can leverage these advancements by sharing molecular pre-trained models that have been trained on proprietary molecular datasets without compromising data privacy. Then, model users are able to query these pre-trained models without access to any training data.

However, such model-sharing collaboration, while highly beneficial, is not devoid of vulnerabilities. One significant risk is the susceptibility to data extraction attacks [6, 7], where those with malicious intentions may attempt to access the private molecular training data. Such breaches could potentially compromise commercial secrets, violate privacy regulations, and undermine trust among collaborative partners [28]. As the first in the literature, this work studies the problem of *molecular data extraction attacks*, aiming to explore the risks of extracting training data from molecular pre-trained models.

Extensive work has been done on data extraction attacks in the realms of image and text, suggesting that training data can be extracted from pre-trained models due to memorization effects [6, 7]. Yet, these methods are not applicable to molecular pre-trained models for three key reasons. *Firstly*, most of the existing data extraction attacks target at generative models (e.g., transformers and diffusion models) in text and image domains. From generative models, training data can be easily inferred via simple prompts (as depicted in Figure 1(a)). In comparison, most molecular pre-trained models are not generative. Instead, users of these models can only query the model with molecular graph and then obtain corresponding representations of this graph [62, 53, 18, 19, 55], which obstructs the direct extraction of molecular data (see Figure 1(b)). *Secondly*, while models in image and text domains typically employ widely recognized architectures like diffusion models [44] and transformers [43], molecular pre-trained models feature a much greater diversity in their architectures and training tasks, such as contrastive learning [47, 62, 53], context prediction [18, 55, 51], and graph reconstruction [19, 45]. These architectures often remain undisclosed to potential adversaries, adding an additional layer of complexity to any attempt at data extraction. *Lastly*, the vast combinatorial possibilities of molecules, estimated to number around $10^{60}$ [38], introduce a level of complexity that necessitates highly efficient and specialized methods for extracting molecular graphs.

In this paper, we first generate molecule candidates by combining a defined template structure, motif banks, and bond connectivity, all within the bounds of established chemical constraints. This serves

as an alternative approach for molecule generation when direct data extraction from non-generative pre-trained models is not feasible. With these molecule candidates, we introduce a novel scoring function to determine whether potential molecules belong to the training data of a pre-trained model. This scoring function is model-independent, making it applicable across various architectures of molecular pre-trained models. To further reduce the search space for molecules and efficiently extract them, we introduce a Molecule Extraction Policy Network for generating those with high-scoring functions and meet the valency rule through reinforcement learning (RL). Extensive experiments break the illusion that sharing molecular pre-trained models, rather than raw data, adequately protects against data extraction attacks: despite only having query access to these black-box models, our findings reveal a significant risk of training data being extracted with an average precision of 49.0%.

## 2 Problem Formulation

In this section, we outline the scenario, describe the adversary's knowledge, and define the problem associated with molecular graph extraction attacks.

**Scenario.** Consider a real-world scenario within the pharmaceutical industry, where two companies collaborate under a commercial arrangement. Company A provides Company B with query access to a molecular pre-trained model to enhance drug research, for which Company B compensates with a usage fee. However, driven by the intent to gain a competitive advantage or reduce development costs, Company B may, with malicious intent, attempt to extract proprietary training data from this model. We explore the risks of data extraction associated with such a model-sharing collaboration, highlighting the potential for misuse and the ethical considerations it raises.

**Adversary's knowledge.** The adversary (in this case Company B) has *black-box* access to the molecular pre-trained model. This access allows the adversary to query the model with a molecular graph and receive the corresponding graph representation in return, without any insight into the model architecture or the specific pre-training tasks it underwent. This setting mirrors common situations in the industry, particularly for models that are accessed via an API while keeping their internal workings undisclosed [2, 23, 27].

Additionally, adversaries may possess an *auxiliary dataset* ($\mathcal{G}_{\text{aux}}$), which can be used to assist with data extraction efforts. This assumption is reasonable given that such an auxiliary dataset can be sourced from publicly available molecular databases like ChEMBL [14], PubChem [51], ZINC15 [46], or it could be some data held by the adversaries themselves.

**Molecular graph extraction attack.** In this paper, we explore the potential risk of molecular pre-trained models leaking their training molecule data. To thoroughly investigate this risk, we formally define the problem as follows:

**Problem 1 (Molecular Graph Extraction Attack)** *Given the molecular pre-trained model $f$ that has been pre-trained on a **private dataset** $\mathcal{G}$, adversaries who only have query access to the model and access to an auxiliary dataset aim to obtain a subset of graphs $\mathcal{G}_{adv}$ that exist within $\mathcal{G}$.*

Here, the private information is defined as the molecular graphs within $\mathcal{G}$. Since we are investigating the risks posed by molecular graph extraction attacks, we consider it sufficiently risky to deduce only a portion of the graphs or those similar to $\mathcal{G}$.

## 3 Methodology

We first introduce the molecule generation process to generate molecule candidates in §3.1. Further, we present a model-independent scoring function in §3.2. Finally, to reduce the vast exploration space, we propose a Molecule Extraction Policy Network in §3.3.

### 3.1 Molecule Generation Process

Since molecular pre-trained models are non-generative, conducting a data extraction attack requires generating potential molecules to query the model. Here we design a molecule generation mechanism that specifically takes into account three key elements of molecule data: template structure, motif banks, and bond connectivity. By defining these elements, we can choose a template structure as a starting point and continuously select motifs and bonds that satisfy biochemical constraints to construct viable molecules.

**Template structure.** Before generating a molecule, we need to establish an effective starting point for the molecule, which is referred to as the template structure. This template structure plays a foundational role and should meet two strategic criteria: (1) Functional: the template structure should constitute the core structure of molecules. This structure influences the molecular properties but does not necessarily determine them. (2) Common: the template structure should be prevalent across a wide range of molecules. Both criteria are indispensable: If a template structure is only common but lacks functional significance, like an atom, then it fails to provide useful information on the structure of the molecules under interest. On the other hand, if a template is merely functional, such as thiols (typically found only in antioxidant molecules) [39], then the diversity of the extracted molecules would be restricted. Rings uniquely meet both criteria: they are prevalent across numerous molecular families, satisfying the common criterion, and as a functional structure, they significantly influence molecular stability, reactivity, and interactions with other molecules [9, 48]. Recognizing these advantages, we select rings as the template structure for molecule generation.

**Motif bank.** After a template structure is selected as the starting point for molecule generation, adversaries can then attach various molecule building blocks. Common choices include atoms [61, 31] and motifs [25, 60]. Yet, atoms might not be informative attachments, due to the limited structural information an individual atom can provide. Moreover, atoms may form atypical chemical fragments, such as alternating bond patterns that form incomplete aromatic rings [33]. Therefore, we opt for motifs as the building blocks for molecule generation. In our implementation, we use the 91 common motifs extracted by [33]. These common motifs constitute the motif bank.

**Bond connectivity.** Once we establish the template structure and motif banks, the next step is to consider the connectivity between the template structure and the motifs. Molecules are generated by forming bonds through specific attachment positions on the template structure and motifs. However, it is often not legal to attach an arbitrary bond to an arbitrary position. So expert chemical knowledge is needed in this process, and common chemical constraints should be satisfied. In this work, we obtain feasible attachment positions by utilizing CReM [37] for the decomposition of molecules in the auxiliary dataset $\mathcal{G}_{\text{aux}}$.

Given a set of template structures and a motif bank, we first select a template structure $R$ as the starting point and choose a motif $M$ from the motif bank. Then, a bond $B = \{a_R, a_M\}$ is formed between the template structure $R$ and the motif $M$, subject to the satisfaction of chemical constraints, where $a_R$ and $a_M$ represent the attachment points in $R$ and $M$ respectively. This bonding results in $\hat{G}$ as the union of $R$ and $M$ with the bond $B$, which can be represented as $\hat{G} := R \underset{B}{\cup} M$. In subsequent generation steps, we can take $\hat{G}$ as our new starting point and select additional motifs and bonds. By repeating this process, we gradually construct a potential molecule.

### 3.2 Scoring Function Design

This subsection introduces a scoring function to determine the probability of the existence of $\hat{G}$ in the private training dataset $\mathcal{G}$. The scoring function should be independent of any specific model architectures or pre-training tasks. In the following, we first define the scoring function, and then explain its rationality.

Since adversaries can only query the molecular pre-trained model to obtain representations, we derive insights from the representations of template structure $R$, the motif $M$, and their combined structure with bond $B$, denoted as $\hat{G}$, as provided by the pre-trained model $f$. We define the scoring function as follows:

$$\text{Score}(R, M, \hat{G}) = \text{Sim}(f(\hat{G}), \alpha f(R) + (1 - \alpha)f(M)), \tag{1}$$

where $\alpha \in [0, 1]$ is a hyper-parameter, $f(\hat{G})$, $f(R)$, and $f(M)$ are representations of $\hat{G}$, $R$, and $M$ respectively, and $\text{Sim}(\cdot, \cdot)$ can be defined as cosine similarity or other forms of similarity measure.

**Rationality of scoring function.** The crux of the scoring function's rationality lies in the observation that, for molecular pre-trained model, the relationship between representations $f(\hat{G})$, $f(R)$, and $f(M)$ exhibits distinct patterns depending on whether $\hat{G}$ is present in the private training dataset.

Consider Figure 2 as an illustrative example. In the top row, if the molecule $\hat{G} := R \underset{B}{\cup} M$ exists in $\mathcal{G}$, the obtained representation of $R$ often contains information about $M$, due to their frequent co-occurrence. Conversely, the obtained representation of $M$ contains information about $R$. Con-

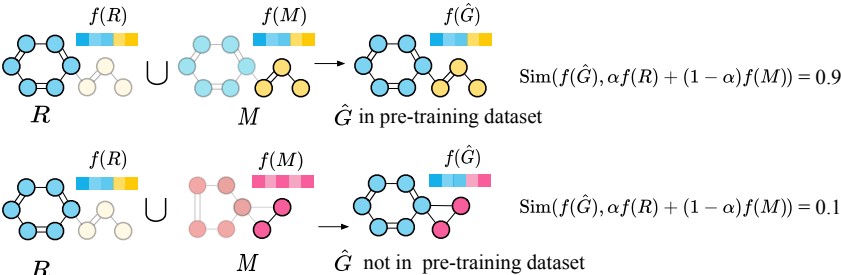

Figure 2: Illustration of the validity in scoring function design. *Top row:* the case that generated molecule $\hat{G}$ exists in the private training dataset $\mathcal{G}$. *Bottom row:* the case that $\hat{G}$ does not exist in $\mathcal{G}$.

sequently, there exists a specific relationship: $f(\hat{G})$ can be effectively approximated as a linear combination of the representations of two other molecules, $f(R)$ and $f(M)$, in the representation space, that is, $f(\hat{G}) \approx \alpha f(R) + (1-\alpha)f(M)$. On the other hand, from the bottom row of this figure, if $\hat{G}$ does not exist in $G$, it is highly likely that $f(\hat{G})$ is dissimilar with $\alpha f(R) + (1-\alpha)f(M)$.

Additionally, we show that the scoring function is related to specific molecular pre-trained models with various $\alpha$ values in Eq. (1). For instance, when the molecular pre-trained model employs bond-deletion augmentation in graph contrastive learning [62, 53], the value of $\alpha$ in Eq. (1) is approximately $\frac{1}{2}$. For the molecular pre-trained model that employs subgraph augmentations in graph contrastive learning [45], the value of $\alpha$ in Eq. (1) is approximately 1. Detailed proofs supporting these examples, as well as rationale behind the scoring function can be found in Appendix A.5. Furthermore, our analysis of the distribution of $\alpha$ values across different pre-trained models in §4.2 provide additional evidence.

**Learning scoring function using auxiliary dataset.** However, directly relying on the scoring function Eq (1) to extract training data still presents two challenges.

Firstly, the graph representations obtained from the pre-trained model may not be optimally suited for data extraction due to the discrepancy between the pre-training tasks and the task of data extraction. To mitigate this issue, we introduce an adapter $g_\theta$, instantiated as an MLP with learnable parameters $\theta$. This adapter is designed to project the representations obtained from the pre-trained model $f$ into another representation space that is more conducive to facilitating a graph extraction attack. The transformation of representations is achieved through the mapping $g_\theta \circ f(\cdot) = g_\theta(f(\cdot))$, with the hope that the output is specifically tailored for the extraction task.

Secondly, treating $\alpha$ as a fixed hyper-parameter in the scoring function introduces challenges in adaptability. A fixed $\alpha$ may not adjust dynamically to different contexts or datasets, potentially limiting the attack's adaptability and leading to suboptimal performance. To overcome this limitation, we introduce a more adaptive mechanism that can optimize $\alpha$ in response to changing contexts, by modeling $\alpha$ as a function of $\alpha = h_\phi([f(R); f(M); f(\hat{G})])$.

Based on the above solutions, we transform the scoring function Eq. (1) into a learnable form:

$$\text{Score}_{\{\theta,\phi\}}(R, M, \hat{G}) = \text{Sim}(g_\theta \circ f(\hat{G}), \alpha g_\theta \circ f(R) + (1-\alpha)g_\theta \circ f(M)). \qquad (2)$$

We utilize the information contained in the auxiliary dataset $\mathcal{G}_{\text{aux}}$ to learn the parameters $\{\theta, \phi\}$. The key idea is that if the scoring function, parameterized by $\{\theta, \phi\}$, can effectively determine the presence of a graph $\hat{G}$ within $\mathcal{G}_{\text{aux}}$, it is likely to generalize to the private training dataset $\mathcal{G}$. The training process can be formalized as follows:

$$\min_{\theta,\phi} \; \mathbb{E}_{\hat{G}}\big[\ell_{\text{CE}}\big(\text{sigmoid}(\text{Score}_{\{\theta,\phi\}}(R, M, \hat{G})), \mathbb{1}_{\{\hat{G} \text{ in } \mathcal{G}_{\text{aux}}\}}\big)\big], \qquad (3)$$

where $\ell_{\text{CE}}$ is the cross-entropy function, $\mathbb{1}_{\{\cdot\}}$ is the indicator function, and $\mathbb{E}_{\hat{G}}$ is the mathematical expectation taken over all the possible generated molecules $\hat{G}$.

### 3.3 Molecule Extraction Policy Network

To conduct the molecular extraction attack, the most straightforward way is to enumerate all possible generated molecules and rank them using a specified scoring function, and then select those with the

highest scores. This approach, inevitably, leads to an exponential increase in complexity due to the vast number of possible combinations.

Given that the molecular generation process involves the iterative selection of template structures and motifs to form bonds, it naturally aligns with the Markov Decision Process (MDP) framework [41], where each decision is based on the current state and leads deterministically to a new state. This sequential decision-making property allows for a structured exploration of the molecular space. We therefore introduce a Molecule Extraction Policy Network for molecular graph extraction attacks through RL, which significantly narrows the search space for molecules. This network strategically guides the selection of motifs and attachment points, focusing on the most promising options. We further detail our design.

**State space.** The state at time step $t$, denoted as $S_t$, is defined as the graph $\hat{G}_t$ generated up to that point. The initial state $\hat{G}_0$ represents the template structure $R$, serving as the starting point for the molecular generation process.

**Action space.** At time step $t$, the RL agent selects a motif $M_t$ from the motif bank and determines the best attachment positions $B_t = \{a_{\hat{G}_{t-1}}, a_{M_t}\}$, resulting in the updated graph $\hat{G}_t := \hat{G}_{t-1} \underset{B_t}{\cup} M_t$. More specifically, the action at step $t$ involves three stages: (1) Selecting attachment position $a_{\hat{G}_{t-1}}$ on $\hat{G}_{t-1}$; (2) Choosing a motif $M_t$ from the motif bank; (3) Selecting attachment position $a_{M_t}$ on $M_t$ to form the bond $B_t$. In summary, the action at step $t$ can be expressed as $A_t = \{a_{\hat{G}_{t-1}}, M_t, a_{M_t}\}$.

**Reward design.** We employ both delayed reward and intermediate reward to guide the molecular generation. For the delayed reward, we instantiate the reward $r$ as the scoring function and extend it over multiple steps as follows:

$$r(S_t, A_t) = \sum_{i=0}^{t-1} \beta_i r(S_i, A_i) = \sum_{i=0}^{t-1} \beta_i \text{Score}_{\{\theta,\phi\}}(\hat{G}_{i-1}, M_{i-1}, \hat{G}_i), \tag{4}$$

where $\beta_i$ represents the weight for combining rewards from different $\hat{G}_i$ on the trajectory. Intuitively, if $\hat{G}_{t-1}$ exists in $\mathcal{G}$, then the generated $\hat{G}_t$ based on $\hat{G}_{t-1}$ is more likely to exist in $\mathcal{G}$. Therefore, we consider the molecule generation process as a whole and accumulate rewards by summation. Additionally, when $t$ is small, the corresponding weight of the reward should be relatively high, whereas when $t$ is large, the weight should be relatively low. Here, we set $\beta_i$ to $0.99^i$.

Regarding intermediate rewards, a positive reward $\delta$ is allocated when the generated molecules do not violate valency rules [36], ensuring that each atom has not exceeded its maximum possible valency. For molecules that fail to pass valency rules, the intermediate rewards are set to zero.

**Policy network.** To enable the RL agent to predict actions effectively, obtaining accurate molecular representations is crucial. We utilize a GNN to learn representations from molecules, a method proven effective for learning molecular representations [10]. We can then obtain representations of attachment position $z(a_{\hat{G}_{t-1}})$ and $z(a_{M_t})$. For graph representations, such as motif representations, we apply sum pooling to derive the graph representation, represented as $z(M_t)$.

Based on the representations, three networks ($\pi_{\text{first}}$, $\pi_{\text{second}}$, and $\pi_{\text{third}}$) are designed to predict the action $A_t = \{a_{\hat{G}_{t-1}}, M_t, a_{M_t}\}$ across three stages. For the first stage, the RL agent selects an attachment position from $\hat{G}_{t-1}$ according to the network $\pi_{\text{first}}$, i.e.,

$$p_t^{\text{first}}(a_{\hat{G}_{t-1}}) = \pi_{\text{first}}(z(a_{\hat{G}_{t-1}})), \tag{5}$$

where $\pi_{\text{first}}$ outputs the probability distribution $p_t^{\text{first}}$ of $a_{\hat{G}_{t-1}}$. We then obtain $a_{\hat{G}_{t-1}}$ by sampling according to the probability $\pi_{\text{first}}$. For the second stage, the RL agent tries to select the motif $M_t$ from the motif bank based on selected $a_{\hat{G}_{t-1}}$, i.e.

$$p_t^{\text{second}}(M_t) = \pi_{\text{second}}\left(\left[z(a_{\hat{G}_{t-1}}) : z(M_t)\right]\right), \tag{6}$$

where $\pi_{\text{second}}$ takes in the representations of attachment position $a_{\hat{G}_{t-1}}$ selected in stage 1 and motif $M_t$, outputs the probability $p_t^{\text{second}}(M_t)$ of selecting motif $M_t$. Finally, given selected $a_{\hat{G}_{t-1}}$ and $M_t$, the agent selects attachment position $a_{M_t}$ in motif $M_t$ as:

$$p_t^{\text{third}}(a_{M_t}) = \pi_{\text{third}}\left(\left[z(a_{\hat{G}_{t-1}}) : z(a_{M_t})\right]\right), \tag{7}$$

where $\pi_{\text{third}}$ outputs the probability distribution of $a_{M_t}$. In the implementation, the three policy networks, $\pi_{\text{first}}$, $\pi_{\text{second}}$, and $\pi_{\text{third}}$, consist of MLP layers with ReLU activations, followed by a softmax layer to predict the probabilities $p^{\text{first}}$, $p^{\text{second}}$, and $p^{\text{third}}$, respectively.

**Policy gradient training.** To enhance the exploration capability of the RL agent in capturing more molecules from $\mathcal{G}$, we leverage the Soft Actor-Critic framework [15]. Soft Actor-Critic integrates the entropy measure of the policy into the reward to promote the exploration of molecular generation. By maximizing entropy, we can obtain molecules with both high scores and diversity. Specifically, the policy network is trained with the objective as follows:

$$\max_{\pi=\{\pi_{\text{first}},\pi_{\text{second}},\pi_{\text{third}}\}} \sum_{i=0}^{t-1} \mathbb{E}_{(S_i, A_i)\sim\rho_\pi} \left[r(S_i, A_i) + \tau\mathcal{H}(\pi(\cdot \mid S_i))\right], \tag{8}$$

where $\mathcal{H}(\pi(\cdot \mid S_i))$ is the entropy measure of the action distribution given the state $S_i$ and $\tau$, known as the temperature parameter, controls the trade-off of exploration for molecules. The detailed modifications to the Soft Actor-Critic optimization can be found in Appendix A.3.

**Reward function initialization and update.** Since the reward function depends on the quality of the scoring function's training, and the scoring function's training, in turn, depends on the quality of the generated graphs, we consider initializing the scoring function for a warm-up phase. We first enumerate all possible molecules constructed by appending a motif to the template structure, and use them as the distribution of the generated graph to pre-train adapters $g_\alpha$ and $h_\phi$ using Eq. (3). During the training process, as the quality of the generated graphs improves, the generated molecules can, in turn, enhance the RL learning framework. Specifically, we adjust the scoring function using the generated molecules as the distribution of the generated graph and training with Eq. (3).

## 4 Experiments

In this section, we evaluate the performance of molecular extraction attacks against different molecular pre-trained models. Besides, we conduct case studies, and runtime analyses to underscore the effectiveness of our approach. Additional results can be found in Appendix A.4.

### 4.1 Experimental Setup

**Dataset.** In our experiment, we used datasets containing 2 million molecules sampled from ZINC15 [46] as the pre-training dataset $\mathcal{G}$, and an additional 20,000 molecules as the auxiliary dataset $\mathcal{G}_{\text{aux}}$. The detailed statistics information is provided in the Appendix A.3.

**Molecular pre-trained models.** We selected the most common and widely used molecular pre-trained models from each category to demonstrate the versatility of the proposed method. These methods include: (1) Contrastive Learning: GraphCL [62], SimGRACE [56] InfoGraph [47]; (2) Graph Reconstruction: GraphMAE [18], AttrMasking [19], EdgePred [16], Mole-BERT [55]; (3) Context Prediction: ContextPred [19], Grover [45]. Notably, the encoder architectures of Mole-BERT and Grover are based on Transformers or BERT structures and all the molecular pre-trained models are trained using the default hyper-parameters specified in original papers.

**Baselines.** Since existing methods are not designed for molecular extraction, we first tailor other methods to fit our setting. We enumerate all the potential molecules (constructed within one or two steps generation) and use metrics to select molecules as the prediction of $\mathcal{G}_{\text{aux}}$. Our baselines can be roughly categorized into two groups: chemical property-based methods and learning-based methods.

For chemical property-based methods, we compare the *QED* score [3], a common estimation of drug-likeness, which predicts the drug-like potential of a molecule. Additionally, the *SA*(Synthetic Accessibility) score is considered to measure the synthetic accessibility and rationality of molecules [11]. We also introduce the *Docking* score[49] as our scoring function baseline to estimate the binding affinity between a ligand (small molecule) and a receptor (protein target). Specifically, we obtain

Table 1: We investigate the performance of molecular extraction results across various molecular pre-trained models, examining different values of $K$ and different types of molecules (constructed in one-step or two-step generation). The notation "/" indicates that the runtime exceeded three days.

| | One Step | | | | Two Step | | | |
|---|---|---|---|---|---|---|---|---|
| | $K = 50$ | | $K = 100$ | | $K = 100$ | | $K = 200$ | |
| | Prec. | FCD | Prec. | FCD | Prec. | FCD | Prec. | FCD |
| Random | 0.05 | 21.77 | 0.09 | 17.99 | 0.09 | 23.18 | 0.07 | 23.20 |
| QED | 0.14 | 23.95 | 0.37 | 21.75 | 0.05 | 23.71 | 0.06 | 23.47 |
| SA | 0.43 | 23.18 | 0.21 | 21.37 | 0.33 | 25.97 | 0.30 | 24.46 |
| FA7 | 0.25 | 19.68 | 0.18 | 18.13 | / | / | / | / |
| PARP-1 | 0.27 | 21.85 | 0.23 | 19.47 | / | / | / | / |
| 5-HT1B | 0.25 | 21.49 | 0.25 | 19.08 | / | / | / | / |
| MLP (GraphCL) | 0.48 | 20.47 | 0.32 | 21.06 | 0.29 | 23.17 | 0.19 | 23.17 |
| Ours (GraphCL) | 0.50 | 19.22 | 0.35 | 19.85 | 0.31 | 23.57 | 0.51 | 23.09 |
| MLP (SimGRACE) | 0.43 | 17.44 | 0.32 | 17.09 | 0.50 | 22.81 | 0.38 | 21.58 |
| Ours (SimGRACE) | 0.53 | 17.79 | 0.34 | 16.68 | 0.55 | 22.40 | 0.50 | 22.75 |
| MLP (InfoGraph) | 0.41 | 18.09 | 0.30 | 17.66 | 0.50 | 25.80 | 0.47 | 25.80 |
| Ours (InfoGraph) | 0.51 | 17.12 | 0.32 | 16.51 | 0.55 | 21.47 | 0.61 | 21.16 |
| MLP (GraphMAE) | 0.37 | 18.09 | 0.36 | 17.46 | 0.54 | 38.41 | 0.37 | 38.40 |
| Ours (GraphMAE) | 0.47 | 17.79 | 0.36 | 17.12 | 0.64 | 38.50 | 0.38 | 38.31 |
| MLP (AttrMasking) | 0.61 | 17.56 | 0.37 | 17.42 | 0.48 | 21.93 | 0.24 | 22.15 |
| Ours (AttrMasking) | 0.61 | 17.20 | 0.39 | 16.49 | **0.72** | 21.39 | **0.76** | 20.86 |
| MLP (EdgePred) | 0.61 | 17.56 | 0.37 | 16.98 | 0.59 | 23.82 | 0.59 | 22.77 |
| Ours (EdgePred) | **0.65** | 16.84 | **0.39** | 16.49 | 0.60 | 21.33 | 0.47 | 21.91 |
| MLP (Mole-BERT) | 0.39 | 18.02 | 0.32 | 17.81 | 0.50 | 33.53 | 0.32 | 33.53 |
| Ours (Mole-BERT) | 0.47 | 17.90 | 0.33 | **16.39** | 0.55 | 30.20 | 0.39 | 30.20 |
| MLP (ContextPred) | 0.39 | 18.57 | 0.36 | 17.20 | 0.60 | 21.32 | 0.38 | 21.66 |
| Ours (ContextPred) | 0.45 | 16.76 | 0.36 | 17.18 | 0.65 | 22.12 | 0.44 | 21.33 |
| MLP (Grover) | 0.25 | 17.32 | 0.22 | 17.09 | 0.29 | 18.96 | 0.24 | 18.99 |
| Ours (Grover) | 0.37 | **16.79** | 0.22 | 16.94 | 0.69 | **18.30** | 0.68 | **18.02** |

three variants [60]: *FA7*, *PARP-1*, and *5-HT1B*. As for learning-based methods, we use an MLP classifier to predict the existence of $\hat{G}$ in $\mathcal{G}$. This classifier is trained by predicting the existence in $\mathcal{G}_{\text{aux}}$ based on the representation of $\hat{G}$. Detailed descriptions of baselines and the implementation of models are provided in the Appendix A.3.

**Metrics.** A molecular graph extraction attack is considered successful if a graph in $\mathcal{G}_{\text{adv}}$ exists in $\mathcal{G}$ or if $\mathcal{G}_{\text{adv}}$ is similar to $\mathcal{G}$. Therefore, assuming the model has generated $\mathcal{G}_{\text{adv}}$ with $K$ molecules, we adopt the following metrics to measure the performance of the extraction attack:

- **Precision** measures the ratio of generated molecules that exist within the $\mathcal{G}$. The larger the precision is, the better the performance of the molecular extraction attack.
- **FCD**, also known as Fréchet ChemNet Distance, offers a distance measure between $\mathcal{G}$ and $\mathcal{G}_{\text{adv}}$. This metric leverages ChemNet [40] to capture the differences in both the chemical and biological properties of the molecules. A lower FCD indicates that $\mathcal{G}$ and $\mathcal{G}_{\text{adv}}$ are similar in terms of chemical and biological properties, suggesting better extraction performance.

## 4.2 Experimental Results

**Molecular extraction results.** Table 1 demonstrates the superior performance of our model over baselines across various molecular pre-trained models. Chemical property-based methods generally underperform, likely due to the infrequency of target properties in pre-trained datasets (e.g., QED). The better performance of SA indicates that molecular stability could be a significant indicator of molecule presence in real datasets. It is evident that our reinforcement learning approach significantly outperforms MLP method on precision and FCD across several pre-trained models, with average improvements of 30.9% and 3.97%, respectively, highlighting the effectiveness of our method.

We can also observe that AttrMasking is the most vulnerable to privacy leakage among the molecular pre-trained models. Furthermore, we have also compared the performance of proposed model under different model frameworks, and it consistently succeeds in molecular graph extraction across various

Table 2: Ablation studies on the performance of molecular extraction results

|  | One Step | | | | Two Step | | | |
| --- | --- | --- | --- | --- | --- | --- | --- | --- |
|  | $K = 50$ | | $K = 100$ | | $K = 100$ | | $K = 200$ | |
|  | Prec. | FCD | Prec. | FCD | Prec. | FCD | Prec. | FCD |
| Ours | 0.50 | 19.22 | **0.35** | 19.85 | **0.31** | 23.57 | **0.51** | 23.09 |
| Ours-RL | **0.56** | 18.67 | 0.35 | **17.52** | 0.30 | **21.28** | 0.40 | **20.73** |
| Ours-SA | 0.30 | 19.18 | 0.25 | 19.17 | 0.24 | 24.02 | 0.29 | 23.58 |
| Ours-adapter | 0.47 | **17.95** | 0.30 | 18.17 | 0.29 | 22.49 | 0.42 | 22.85 |
| Ours-hard | 0.43 | 18.48 | 0.29 | 18.28 | 0.28 | 21.92 | 0.35 | 21.42 |

pre-trained model architectures, including those based on BERT or transformers.

**Ablation study.** To validate the effectiveness of each component, ablation studies are conducted on: (1) Ours-RL, which adopts enumeration instead of an explorative RL framework. (2) Ours-SA, where the reward function is replaced with the most effective chemical property-based SA shown in Table 1. (3) Ours-adapter, which calculates the scoring function without adapters outlined in Eq.(1). In addition, we consider using an auxiliary dataset $\mathcal{G}_{\mathrm{aux}}$ that has lower similarity (*i.e.*, higher FCD) to the pre-training dataset $\mathcal{G}$ in order to simulate a more challenging molecular graph extraction attack scenario, and we denote it as Ours-hard.

As shown in Table 2, the superior performance of Ours compared to Ours-RL, Ours-SA, and Ours-adapter highlights the indispensable roles of the reinforcement learning framework, the scoring function, and the adapter for computing the scoring function. The degraded performance of Ours-hard can be attributed to training the scoring function via $\mathcal{G}_{\mathrm{aux}}$ with lower similarity, which in turn lowers the generalizability of the scoring function. However, Ours-hard still exhibits comparable performance and shows the robustness.

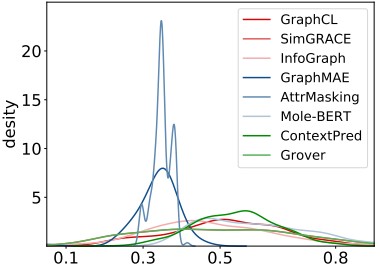

Figure 3: Visualization of $\alpha$ distribution under different pre-trained models. Models in the same category are assigned similar colors for distinction.

Table 3: Comparison of runtime (in seconds) for generating 200 molecules using a 2-step generation process. The learning-based methods are based on the GraphCL molecular pre-trained model.

|  | QED | SA | MLP | Ours |
| --- | --- | --- | --- | --- |
| Score calculation | 632 | 259 | 2,000 | 944 |
| Total | 3,003 | 2,710 | 21,342 | 14,160 |

**Case study.** We further investigate the behavior of the scoring function under various molecular pre-trained models. We explore the $\alpha$ distribution as shown in Figure 3. It is evident that self-supervised tasks within the same category exhibit a similar pattern in their $\alpha$ distributions, whereas models from different categories display distinct distribution patterns. For graph reconstruction-based models, the $\alpha$ distributions are predominantly centered around 0.35, with a peak indicating a high concentration. In contrast, for contrastive learning-based models, the $\alpha$ distributions are flat, which may be attributed to the inherent randomness in the augmentations used in contrastive learning. As for context prediction tasks, the $\alpha$ distributions are centered around 0.54. This phenomenon provides an explanation for the rationality of the scoring function in black-box scenarios.

**Runtime analysis.** Table 3 compares the runtime of the proposed method with the baselines across the two categories: chemical property-based methods, and learning-based methods. The proposed method exhibits superior extraction attack performance while maintaining a runtime that is comparable to others. This efficiency is achieved through the explorative RL framework, which replaces the need

for exhaustive enumeration of thousands of molecules, significantly reducing the time required for molecular graph extraction attacks.

# 5    Related Works

**Molecular pre-trained models**. Molecular pre-trained models utilize GNNs to capture the intricate non-Euclidean structure of molecular graphs, employing various self-supervised pre-training tasks to enhance generalization [57, 22, 12, 5, 21, 58]. Training on extensive molecular graphs, molecular pre-trained models can acquire generalized molecular graph representations and patterns, thereby benefiting various downstream tasks in the molecular domain [13, 53, 1, 30, 32]. Molecular pre-trained models typically employ self-supervised tasks as follows. (1) Contrastive Learning [47, 62, 53]. The objective of the contrastive pre-training task is to capture the similarities and dissimilarities between instances of subgraphs at the molecular level or motif level. (2) Graph Reconstruction [18, 55, 51]. Certain components (such as atoms, bonds, properties of atoms, and fragments) of molecules are masked out, and models are trained to recover components based on the remaining information. (3) Context Prediction [19, 45]. The objective of graph context prediction is to utilize subgraphs to make predictions of surrounding graph structures. This is achieved by classifying whether a specific neighborhood component and surrounding context belong to the same node within the ego-graph.

**Data extraction attacks.** Effectiveness and reliability of model can be compromised by adversarial attacks in various forms [63, 58]. Pre-trained models contain a large amount of knowledge, and data extraction attacks are among the methods aimed at extracting training data from these models [4, 24]. Research in this area can be broadly classified into two categories: one uses membership inference to deduce information from generative models, while the other exploits the memorization mechanism of networks to carry out attacks. In the first category, [6] generates text from pre-trained language models and performs membership inference attacks to filter the generated text for extraction. In the second category, [26] demonstrated that the effectiveness of data extraction is due to duplication in commonly used web-scraped training sets [26]. [20] analyzed the extracted text from pre-trained language models and found these models do leak personal information as a result of memorization. However, all the aforementioned studies focus solely on the extraction from generative pre-trained models and do not adequately address the challenge of extracting data from graph pre-trained models.

# 6    Broader Impacts

We recognize that our investigation into Molecular Graph Extraction Attacks on graph-pretrained models could be misused, particularly in collaborative model-sharing, where it may lead to privacy risks. However, we emphasize that our primary objective is to identify vulnerabilities in graph pretrained models, and support the creation of more effective defense strategies. To this end, the paper assesses the susceptibility of mainstream graph-pretrained models to the attack, underscoring the need for enhanced defense measures for existing work.

Furthermore, we propose the following potential defense strategies: (1) Behavior Detection: Implement systems for continuous monitoring and identification of malicious queries in shared models to protect data integrity. (2) Prediction Perturbation: Since the efficacy of model extraction attacks is influenced by embeddings, we suggest introducing minor noise into the final outputs of graph-pretrained models without significantly affecting performance. We believe this ongoing interplay between attack and defense will foster a more robust research community, contributing to future studies on defense strategies.

# 7    Conclusion

The presented work, for the first time, aims to extract private training data from molecular pre-trained models. More specifically, we propose a reinforcement learning framework for molecule graph extraction attacks. We introduce a molecule generation approach and propose a well-motivated scoring function for selection. Experiments show that our proposed framework and scoring function can effectively perform the molecule extraction attack.

**Acknowledgments**

This work was partially supported by National Natural Science Foundation of China (No. 62206056, No. 92270121, No. 62176233, No. 62441605), CIPSC-SMP-Zhipu Large Model Cross-Disciplinary Fund (ZPCG20241030332) and the Fundamental Research Funds for the Central Universities.

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

# A Appendix

## A.1 Notations

The main notations can be found in the following table.

| Notation | Description |
|---|---|
| $\mathcal{G}, \mathcal{G}_{\text{aux}}, \mathcal{G}_{\text{adv}}$ | private graph sets, auxiliary graph sets, graph set extracted from the pre-trained model |
| $f, f(G)$ | molecular pre-trained model, pre-trained representation of graph $G$ |
| $R, M, \hat{G}$ | template structure, motif, generated graph |
| $B, a_R, a_M$ | bond between $R$ and $M$, attachment points in $R$ and $M$ |
| $\alpha$ | parameter in the scoring function |
| $g_\theta, h_\phi$ | adapter for mapping pre-trained representations to perform data extraction with learnable parameters $\theta$ and $\phi$ |
| $\mathbb{P}(\hat{G})$ | the distribution of generated graph $\hat{G}$ |
| $S_t, A_t$ | state, action at time step $t$ |
| $r$ | reward function |
| $\pi_{\text{first}}, \pi_{\text{second}}, \pi_{\text{third}}$ | policy network |
| $\mathcal{H}, \tau$ | entropy measure, temperature parameter |
| $z$ | embedding obtained by policy network |
| $\delta$ | intermediate reward |

Table 4: Description of major notations.

## A.2 Framework

In this section, we detail the pseudocode for the algorithm behind ours. the

---
**Algorithm 1** Algorithm of the proposed model

---
**Require:** Template structure $R$, motif $M$ in motif bank, auxiliary dataset $\mathcal{G}_{\text{aux}}$, and budget $K$.
**Ensure:** extracted graph sets $\mathcal{G}_{\text{adv}}$.
 1: Enumerate all possible molecules constructed by appending a motif to the template structure.
 2: Training the scoring function in Eq. (3) with above generated molecules.
 3: Initialize parameter for RL environment and initial reward function with scoring function.
 4: **for** each iteration **do**
 5:    **for** each environment setup **do**
 6:       Obtain $A_t = \{a_{\hat{G}_{t-1}}, M_t, a_{M_t}\}$ by sequentially compute Eq. (5), Eq. (6) and Eq. (7).
 7:       $\hat{G}_t := \hat{G}_{t-1} \underset{B_t}{\cup} M_t$
 8:       Compute the overall reward $r(S_t, A_t)$ and intermediate reward.
 9:       Optimization for Soft Actor-Critic.
10:    **end for**
11: **end for**
12: Utilizing the trained reinforcement learning agent, generate $K$ molecules for $\mathcal{G}_{\text{adv}}$.

---

## A.3 Addition Experimental Setup

**The detailed statistics of the pre-training dataset and auxiliary dataset.** The pre-training dataset consists of 1,883,524 molecules, while the auxiliary dataset contains 20,000 molecules. There is an overlap of 103 molecules between these datasets, indicating a relatively small overlap ratio.

To enhance computational efficiency, we subdivide both the pre-training and auxiliary datasets based on template structures. For each template structure, we select molecules containing the corresponding template from the original dataset to create a tailored dataset. Subsequently, we perform attacks and evaluations using these customized pre-training and auxiliary datasets.

Additionally, to evaluate the robustness of our attack method, we utilize a set of hard auxiliary datasets. For a given template structure, we select 80% of the molecules from the auxiliary dataset to form a hard auxiliary dataset. This hard auxiliary dataset has a higher Fréchet ChemNet Distance (FCD) from the pre-training dataset compared to the original, resulting in reduced similarity and increased difficulty of the attack. Below are the statistics for some of these datasets:

| Index of template structure | 0 | 1 | 2 |
|---|---|---|---|
| # of molecules in $\mathcal{G}$ | 18 | 59 | 33 |
| # of molecules in $\mathcal{G}_{\text{aux}}$ | 59 | 15 | 31 |
| FCD with $\mathcal{G}$ | 46.15 | 30.41 | 38.67 |
| # of molecules in hard $\mathcal{G}_{\text{aux}}$ | 47 | 12 | 24 |
| FCD with $\mathcal{G}$ | 47.21 | 33.71 | 40.23 |

**Baseline model description.** The models we have chosen include:

- **GraphCL** [62] is a contrastive self-supervised learning method for GNNs, which learns representations by maximizing the agreement between differently augmented views of the same graph.

- **SimGRACE** [56] utilizes the original graph as input, and employs a GNN model along with its perturbed variant as dual encoders. Then model conducts two correspondingly linked perspectives for contrastive learning without the need for data augmentation.

- **InfoGraph**[50] is a pre-training approach that maximizes the mutual information between the local patch representations and global graph representations, encouraging the model to capture local and global graph structures.

- **GraphMAE** [18] is a graph-based model that uses a masked autoencoder framework for pretraining, which learns to reconstruct masked parts of input graphs, enabling the model to capture intrinsic graph structures.

- **Attribute Masking** [19] aims to capture domain knowledge by learning the regularities of the node/edge attributes distributed over graph structure.

- **EdgePred** [16] is a pre-training task where the model learns to predict whether an edge (relationship) exists between two nodes in a graph, which helps the model to understand the connectivity and relationship between nodes.

- **Mole-BERT** [55] utilizes a variant of VQ-VAE as a context-aware tokenizer to encode atom attributes and introduces a new node-level pre-training task, Masked Atoms Modeling along with triplet masked contrastive learning (TMCL) for graph-level pre-training

- **ContextPred** [19] is a pre-training task where the model learns to predict the context of a given node or subgraph.

- **Grover** [45] learns molecular representations by predicting the original training task after self-supervised pre-training.

**Description of baselines.** We compare with the following baselines.

- **MLP** trains a multi-layer classifier to predict the graph $\hat{G}$'s existence in $\mathcal{G}$. The classifier takes in $\hat{G}$'s representation $f(\hat{G})$, and is trained to predict $\hat{G}$'s existence in $\mathcal{G}_{\text{aux}}$.

- **QED score** [3] stands for quantitative estimation of drug-likeness, predicting the drug-like potential of a molecule. The higher the QED score, the more likely the molecule is a drug. Based on the assumption that a drug-like molecule is likely to appear in $\mathcal{G}$, a potential molecule with a high QED, is predicted to exist in the pre-training dataset.

- **SA score** [11] stands for synthetic accessibility score, estimating the ease of synthesizing a particular molecule. SA scores typically range from 1 (easily synthesizable) to 10 (difficult to synthesize), with a lower SA score indicating that the molecule is easier to synthesize. Therefore, we use the negative SA score to score molecules.

- **Docking score** [49] estimates the binding affinity between a ligand (small molecule) and a receptor (protein target). A more negative docking score indicates a stronger binding interaction between the

ligand and the receptor, implying a more favorable binding event. Therefore, we use the negative docking score to score the potential molecule $\hat{G}$. Specifically, we get 3 variants of docking scores with three different protein targets **FA7**, **PARP-1**, **5-HT1B**.

**Detailed implementation details.** Since existing methods are not tailored for molecular extraction attacks, we initially adapt other methods to our specific scenario. We enumerate all potential molecules (constructed through either one-step or two-step generation) and employ metrics to select molecules as predictions for $\mathcal{G}_{\text{aux}}$.

For chemical property-based methods, we utilize the rdkit toolkit [29] for computations. Meanwhile, for learning-based methods, we employ a two-layer MLP classifier with a hidden size of 300.

In constructing our reinforcement learning framework, we employ the Soft Actor-Critic (SAC) algorithm as implemented in OpenAI's SpinningUp, as well as the molecular reinforcement learning efforts [60]. Regarding the parameters for the RL agent, we set the total number of training epochs to 100, with $\beta_i$ as the weight for delayed reward and $\delta = 0.05$ for intermediate rewards. The graph adapter within the scoring function of the delayed reward is also a two-layer MLP with a hidden size of 300. We employ the Adam optimizer with a learning rate of 0.005 for 100 epochs during the pre-training phase. After 5 epochs, the scoring function is trained using the generated molecules. For policy training, we implement three policy networks with two-layer MLPs and a hidden size of 128. The graph representation network utilizes a two-layer GCN with a hidden size of 128. We update the policy network after generating 256 molecules, and set the temperature $\tau$ to 1. The policy networks are trained with the Adam optimizer, using a learning rate of 0.01 and a weight decay of 1e-6. All experiments are conducted on a single machine of Linux system with an Intel Xeon Gold 5118 (128G memory) and a GeForce GTX Tesla P4 (8GB memory).

**Modification to the optimization Soft Actor-Critic.** Similar to the optimization process in [15], calculating entropy requires computing $\log \pi(A_i \mid S_i)$. In our setup, the specific method for this calculation is as follows:

$$
\begin{aligned}
\log \pi(A_i \mid S_i) &= \log \pi(a_{\hat{G}_{i-1}}, M_i, a_{M_i} \mid \hat{G}_i) \\
&= \log \pi_{\text{first}}(a_{\hat{G}_{i-1}} \mid \hat{G}_i) + \log \pi_{\text{second}}(M_i \mid \hat{G}_i, a_{\hat{G}_{i-1}}) + \log \pi_{\text{third}}(a_{M_i} \mid M_i, a_{\hat{G}_{i-1}})
\end{aligned}
\tag{9}
$$

## A.4  Additional Experimental Results

**Rationality behind ring structures.** We here clarify rationality behind choosing ring structures as a starting template is that ring structures are very common in chemical datasets. In the ZINC15 dataset with 2 million unlabeled molecules, we identified 1,990,890 molecules containing ring structures, constituting the majority of the dataset. Furthermore, the diversity of ring structures offers a wide range of options for template structures (such as tetrahydrofuran and cyclobutane).

We further incorporate five ring-free scaffold templates (i.e., alkanes) into the template bank. The performance is shown as follows. It can be observed that incorporating these ring-free templates has led to a slight improvement in performance.

Table 5: Ablation studies on the performance of molecular extraction results

|  | One Step | | Two Step | |
| --- | --- | --- | --- | --- |
|  | $K = 50$ | $K = 100$ | $K = 100$ | $K = 200$ |
|  | Prec. FCD | Prec. FCD | Prec. FCD | Prec. FCD |
| Ours | 0.50 19.22 | 0.35 **19.85** | 0.31 23.57 | 0.51 **23.09** |
| Ours-ring | **0.51 19.11** | **0.36** 20.18 | **0.37 23.27** | **0.52** 23.18 |

**Result for multiple step generation.** Our approach based on reinforcement learning can be extended to multiple steps. Here, we extend the time-step, and the results are as follows (we only take "Random" as a baseline considering the runtime of other baselines). It can be observed that as the time-step increases, performance may decline due to the increased difficulty of extraction caused by the complexity of the molecules. Nevertheless, our model still outperforms the baseline model in precision, indicating the effectiveness of our extraction method.

Table 6: Precision of molecular extraction results among different time step

|  | One Step | | Two Step | | Three Step | | Four Step | |
|---|---|---|---|---|---|---|---|---|
|  | $K = 50$ | $K = 100$ | $K = 100$ | $K = 200$ | $K = 100$ | $K = 200$ | $K = 100$ | $K = 200$ |
| Random | 0.05 | 0.09 | 0.09 | 0.07 | 0.02 | 0.00 | 0.00 | 0.00 |
| Ours | **0.50** | **0.35** | **0.31** | **0.52** | **0.27** | **0.25** | **0.17** | **0.18** |

**Performance with PebChem as the auxiliary dataset.** We also sampled 20,000 molecules from PubChem [51] as the auxiliary dataset to ensure a difference from the ZINC pre-training dataset. The results based on the GraphCL pre-trained model are shown as follows. We observed that under these scenarios, the performance of our method had a slight decline. However, it still demonstrates comparable efficacy and showcases robustness.

Table 7: Performance of molecular extraction results with PebChem as the auxiliary dataset.

|  | One Step | | | | Two Step | | | |
|---|---|---|---|---|---|---|---|---|
|  | $K = 50$ | | $K = 100$ | | $K = 100$ | | $K = 200$ | |
|  | Prec. | FCD | Prec. | FCD | Prec. | FCD | Prec. | FCD |
| Ours | 0.50 | 19.22 | **0.35** | 19.85 | **0.31** | 23.57 | **0.51** | **23.09** |
| Ours-PubChem | **0.52** | **18.52** | 0.33 | **17.90** | 0.26 | **20.57** | 0.31 | 24.41 |

**Performance under regression task.** We further explored data extraction attacks on regression models. When integrating regression tasks, we adapted our model by replacing the final output of the regression with the representation in Eq. (2). In our implementation, we chose the real-world chemical dataset FreeSolv [35] and regressed the hydration-free energy, utilizing 5% of the molecules from FreeSolv as an auxiliary dataset. The detailed results are as follows. We discovered that our model still performs well with the regression model.

Table 8: Performance of molecular extraction results under regression task.

|  | One Step | | | | Two Step | | | |
|---|---|---|---|---|---|---|---|---|
|  | $K = 50$ | | $K = 100$ | | $K = 100$ | | $K = 200$ | |
|  | Prec. | FCD | Prec. | FCD | Prec. | FCD | Prec. | FCD |
| MLP | 0.39 | 19.00 | 0.21 | **16.82** | 0.22 | **17.26** | 0.16 | 17.30 |
| Ours | **0.39** | **17.38** | **0.28** | 17.42 | **0.29** | 18.31 | **0.33** | **16.63** |

## A.5 Proofs

Here, we provide a detailed explanation of the example presented in § 3.2 along with its proof, which further indicates that our scoring function can effectively characterize different molecular pre-training tasks.

**Example 1** *When the pre-trained model performs specific subgraph masking as described in [45], the value of $\alpha$ in Eq. (1) is approximately 1. When the pre-trained model is involved in using bond-deletion augmentation in graph contrastive learning, as in [62, 53], the value of $\alpha$ in Eq. (1) is approximately 0.5.*

**Proof for Example.** Given pre-trained model $f$, template structure $R$, motif $M$ and generated moleculars $\hat{G} := R \underset{B}{\cup} M$. Assume that the GNN architecture all adopts the mean pooling as graph pooling. Denote $|\cdot|$ represents the number of nodes, and $f(\cdot)$ denotes the representation, and $\hat{f}(S)$ denotes the representation output by the mean pooling of subset $S$ in $\hat{G}$. Therefore we have:

$$f(\hat{G}) = \frac{|R|}{|R| + |M|}\hat{f}(R) + \frac{|M|}{|R| + |M|}\hat{f}(M) \tag{10}$$

When a pre-trained model employs subgraph masking, it can ensure that the representation of the graph with masked motifs remains consistent with the original graph's representation. That is, $f(\hat{G}) = f(R)$. In this case, the form of our scoring function is as follows:

$$\text{Score}(R, M, \hat{G}) = \text{Sim}\left(\frac{|R|}{|R| + |M|}\hat{f}(R) + \frac{|M|}{|R| + |M|}\hat{f}(M), \frac{|R|\alpha}{|R| + |M|}f(R) + \frac{|M|(1 - \alpha)}{|C| + |M|}f(M)\right)$$

(11)

By combining Eq. (10) and Eq. (11), we have:

$$\text{Score}(R, M, \hat{G}) = \text{Sim}(\frac{|R|}{|R| + |M|}\hat{f}(R) + \frac{|M|}{|R| + |M|}\hat{f}(M), \frac{|R|^2\alpha}{(|R| + |M|)^2}\hat{f}(R)$$
$$+ \frac{|R||M|\alpha}{(|R| + |M|)^2}\hat{f}(M) + \frac{|M|(1 - \alpha)}{|C| + |M|}\hat{f}(M))$$

(12)

By adjusting $\alpha$ in Eq. (12), we can observe that the scoring function reaches its maximum when $\alpha = 1$.

When a pre-trained model is involved in using bond-deletion augmentation in graph contrastive learning, where the representation of the graph with the bond between $M$ and $R$ removed is made similar, it is easy to obtain the following relationship: $\hat{f}(R) \approx f(R), \hat{f}(M) \approx f(M)$. By substituting it into Eq. (11), we can obtain the following results:

$$\text{Score}(R, M, \hat{G}) = \text{Sim}\left(\frac{|R|}{|R| + |M|}f(R) + \frac{|M|}{|R| + |M|}f(M), \frac{|R|\alpha}{|R| + |M|}f(R) + \frac{|M|(1 - \alpha)}{|C| + |M|}f(M)\right)$$

(13)

Apparently, the scoring function reaches its maximum value when $\alpha = 0.5$.

**Proof for the rationale behind the scoring function.** Assume that graph $G$ is composed by $G := G_1 \cup G_2$ and the graph pre-trained model is represented by $f$. Let the loss function for the pre-training task be $\mathcal{L}$, which takes graph representation as input. Further, we assume that $\mathcal{L}$ is a bijection and linear mapping.

Without loss of generality, we assume that the loss function belongs to the category of weighted sum, that is $\mathcal{L}(f(G)) = \alpha_1 \mathcal{L}(f(G_1)) + \alpha_2 \mathcal{L}(f(G_2))$, with $\alpha_1$ and $\alpha_2$ serve as hyper-parameters (This assumption is common among various tasks. For instance, in the most common case of cross-entropy for the classification task, $\alpha_1 = |G_1|/|G|$ and $\alpha_2 = |G_2|/|G|$.). We can infer that $f(G) = \mathcal{L}^{-1}(\alpha_1 \mathcal{L}(f(G_1)) + \alpha_2 \mathcal{L}(f(G_2))) = \alpha_1 f(G_1) + \alpha_2 f(G_2)$.

Therefore, given the theoretical analysis, we can observe that the relationship between $f(G)$, $f(G_1)$, and $f(G_2)$ is akin to a weighted combination, which justifies our design of score function.

