# OpenReview forum: "Extracting Training Data from Molecular Pre-trained Models"
_NeurIPS.cc/2024/Conference — NeurIPS 2024 poster_

### Official Review · Reviewer_qKJK · 2024-07-10

**Soundness:** 3
**Presentation:** 3
**Contribution:** 2
**Rating:** 3
**Confidence:** 4

**Summary:**

This paper tackles the problem of extracting private training molecular data from pre-trained models. To address this problem, the authors propose a machine learning method based on a model-independent scoring function and a molecule extraction policy network. The privacy of the training data is an emerging issue in scientific applications, and this paper showed that the proposed method has practical potential in the problem of extracting private training molecular data.

**Strengths:**

1. Well-organized manuscript.
2. Clear motivation and problem definition.
3. Comprehensive experiments to demonstrate the effectiveness of the proposed method.

**Weaknesses:**

1. Although the authors made a scenario to clarify the problem, it is not realistic because we can easily hide the molecular representations of classification and regression models.
---
2. If the authors claim the importance of the training data privacy in the molecular representation learning tasks, I cannot agree on it because extensive molecular structures for representation learning are already available in public databases, such as PubChem and ChEMBL. Note that the molecular data for supervised learning, rather than representation learning, is expensive because measuring the target physical and chemical properties of the molecules is time-consuming.
---
3. The score function in Eq. (1) is trivial, and the manuscript does not present some theoretical or chemical justification for the score function.
---
4. The proposed method should be evaluated in the regression tasks to demonstrate its practical potential in real-world chemical applications.

**Questions:**

Please refer to the Weaknesses section.

**Limitations:**

Please refer to the Weaknesses section.

---

> ### Author Rebuttal · Authors · 2024-08-07
>
> ### Response to Reviewer qKJK.
>
> We greatly appreciate reviewer qKJK’s insightful feedback and critical comments to help us refine our work.
>
> >W1\&4: Although the authors made a scenario to clarify the problem, it is not realistic ...; The proposed method should be evaluated in the regression tasks ...
>
> Thanks for your comment. We first clarify that our research is centered on model-sharing collaboration for molecular data. In this context, sharing a graph representation learning model is more realistic than sharing classification or regression models due to several reasons. On one hand, from the perspective of data owner, sharing a classification/regression model poses a higher privacy risk due to the potential leakage of confidential label information, such as properties of molecules. These labels are highly valuable and sensitive, as highlighted by your insightful comment, making data owners hesitant to share models trained on such confidential data. Under such consideration, research efforts have proposed collaborative approaches based on sharing graph representation learning models [1,7]. On the other hand, from the model user's standpoint, there is often a need for a general pre-trained model to facilitate various downstream tasks, e.g., property prediction, drug-target interactions and etc. Under this consideration, since it is impractical for data owner to share different classification/regression models for different tasks, sharing a general representation learning model presents more beneficial. Previous efforts also highlight the necessity for such a general pre-trained representation learning model [2, 3].
>
> We also agree with you that in some target-specific collaboration cases, classification/regression model is shared. Therefore, as suggested, we further explored data extraction attacks on regression models. When integrating regression tasks, we adapted our model by replacing the final output of the regression with the representation in Eq (2). In our implementation, we chose the real-world chemical dataset FreeSolv [4] regressed the hydration free energy, utilizing 5\% of the molecules from FreeSolv as an auxiliary dataset. The detailed results are as follows. We discovered that, our model still performs well with regression model.
>
> |~|One Step||||Two Step||||
> |-|-|-|-|-|-|-|-|-|
> |~|K = 50||K = 100||K = 100||K = 200||
> |~|Prec.|FCD|Prec.|FCD|Prec.|FCD|Prec.|FCD|
> |MLP|0.39|19.00|0.21|**16.82**|0.22|**17.26**|0.16|17.30|
> |Ours|**0.39**|**17.38**|**0.28**|17.42|**0.29**|18.31|**0.33**|**16.63**|
>
> >W2: If the authors claim the importance of the training data privacy in the molecular representation learning tasks, I cannot agree on ...
>
> We would like to clarify that privacy protection of molecular structures is also significant.
> Take a real-world example, consider the MELLODDY (MachinE Learning Ledger Orchestration for Drug Discovery) project, where 10 pharmaceutical companies collaborated to train a model for structure-activity. These companies had substantial privacy concerns regarding their molecule structure, treating them as confidential information that cannot be shared [6]. In addition, previous research has demonstrated that molecular structure often involves sensitive intellectual property information and must be controlled by the owning company at all times [4]. Moreover, information on the structural similarity between partners’ compounds is also considered sensitive [5]. These underscore the privacy issue of molecular structures.
>
> We also thank the reviewer reminding us that the labels for molecular data, such as target physical and chemical properties, are also private. So we extend our model's application to share classification/regression model  that is trained on labels (see our response to W1). We will include these discussions in the revised version and explore this aspect further in the future. Thank you for providing this important perspective again.
>
> >W3: The score function in Eq. (1) is trivial...
>
> Thank you for your valuable and insightful feedback. Here, we illustrate the rationale behind the scoring function from a theoretical perspective.
>
> Assume that graph $G$ is composed by $G:=G_{1} \cup G_{2}$ and graph pre-trained model is represented by $f$. Let the loss function for the pre-training task be $\mathcal{L}$, which takes graph representation as input. Further, we assume that $\mathcal{L}$ is a bijection and linear mapping. Without loss of generality, we assume that loss function belongs to the category of weighted sum, that is $\mathcal{L}(f(G)) = \alpha_1 \mathcal{L}(f(G_1)) + \alpha_2 \mathcal{L}(f(G_2))$, with $\alpha_1$ and $\alpha_2$ serve as hyper-parameters (This assumption is common among various tasks. For instance, in the most common case of cross-entropy for classification task, $\alpha_1 = |G_1|/|G|$ and $\alpha_2 = |G_2|/|G|$.).
> We can infer that $f(G)=\mathcal{L}^{-1}(\alpha_1 \mathcal{L}(f(G_1)) + \alpha_2 \mathcal{L}(f(G_2)))=\alpha_{1}f(G_{1})+\alpha_{2}f(G_{2})$.
>
> Given the theoretical analysis, we can observe that the relationship between $f(G)$, $f(G_{1})$, and $f(G_{2})$ is akin to a weighted combination, which justifies our design of score function.
>
> Reference:
>
> [1]  Xie, Han, et al. "Federated graph classification over non-iid graphs." NeurIPS, 2021.
>
> [2] Hu, Weihua, et al. "Strategies for pre-training graph neural networks." ICLR, 2019.
>
> [3] Xia, Jun, et al. "A systematic survey of chemical pre-trained models." IJCAI, 2023.
>
> [4] Mobley, David L., and J. Peter Guthrie. "FreeSolv: a database of experimental and calculated hydration free energies, with input files." 2014.
>
> [5] Simm, Jaak, et al. “Splitting chemical structure data sets for federated privacy-preserving machine learning.”, 2021.
>
> [6] MELLODDY: Machine Learning Ledger Orchestration for Drug Discovery. https://www.melloddy.eu/.
>
> [7] Tan, Yue, et al. "Federated learning on non-iid graphs via structural knowledge sharing." 2023.

---

> ### Author Response · Authors · 2024-08-11
> **Any unanswered questions yet?**
>
> Dear Reviewer qKJK,
>
> Thanks again for your detailed review and questions. We have provided detailed answers to them in the rebuttal. As we are approaching the end of the discussion phase, we would like to kindly ask if you still have any unanswered questions about our paper?
>
> Best, Authors

---

> ### Comment · Reviewer_qKJK · 2024-08-13
>
> Thank you for the careful response.
>
> However, as a researcher in chemical science, I still do not agree with the scenario of this work.
> Researchers and engineers in academic and industrial chemistry want to build a classification or regression model for their molecular datasets because they need to skip time-consuming chemical experiments to observe the physical and chemical properties of target molecules.
>
> The graph representation of the molecule is just intermediate information for classification and regression in most real-world chemical applications. Furthermore, as I mentioned, many large molecular databases containing extensive 2D and 3D molecular structures are publicly accessible, such as PubChem, ChEMBL, and QM. For this reason, training data for molecular representation learning is not expensive and private.
>
> The authors need to handle the validity of their problem definition before developing the method.

---

> > ### Author Response · Authors · 2024-08-13
> > **Response to Reviewer qKJK.**
> >
> > We appreciate your perspective in chemical science and understand the concerns you have raised. Firstly, we recognize the importance of developing classification or regression models in reducing the need for time-consuming experiments. Correspondingly, we have conducted experiments on the FreeSolv regression task and achieved promising results, indicating that our approach is also suitable for regression-type models.
> >
> > Secondly, we believe that due to the confidential nature of label information, and in order to ensure the transferability of the model when dealing with Out-of-Distribution data, selecting to release the intermediate layer representations of the classification/regression model is a reasonable approach in a practical scenario. Specifically, our attack on graph representation is suitable for the scenario.
> >
> > Moreover, when publicly available datasets’ distribution diverge from private data, and the privacy concerns of data owners who are unwilling to disclose any information about the labels, the choice would still be to employ SSL pre-training to obtain graph representations.
> >
> > Lastly, we will revise the problem formulation of our manuscript and incorporate the three cases discussed above. In conclusion, we are committed to refining our manuscript to better align with the concerns and expectations of the chemical science research community. We believe that our work can make a significant contribution to the field, and we are grateful for the opportunity to improve our submission based on your valuable feedback.

---

### Official Review · Reviewer_o6iJ · 2024-07-11

**Soundness:** 3
**Presentation:** 2
**Contribution:** 3
**Rating:** 6
**Confidence:** 2

**Summary:**

This paper explores the vulnerabilities of molecular pre-trained models to data extraction attacks. The authors introduce a novel molecule generation approach and a model-independent scoring function to identify molecules potentially originating from private datasets. They also present a Molecule Extraction Policy Network to optimize the search process for high-scoring molecules.

**Strengths:**

1. The introduction of a novel approach to generate molecules and a scoring function specific to molecular data extraction offers a unique perspective on security concerns associated with pre-trained models.
2. The paper presents a sophisticated technical framework, including reinforcement learning to refine the search for molecular candidates.
3. The availability of the codebase and datasets enhances the reproducibility of the research and facilitates further investigation by the community.

**Weaknesses:**

1. The experiments might not fully capture the diversity and complexity of real-world datasets, which could affect the generalizability of the findings. I recommend conducting additional experiments using a wider variety of datasets to better evaluate the model's performance in real-world scenarios.
2. The scoring function is based on G, which is a linear combination of the representations of R and M. However, considering that the target model is treated as a black-box model, it is unclear whether there is a linear relationship between the representations of R and M obtained by the black-box model and the representation of G. To clarify this point, I suggest providing a thorough analysis of the relationship between the representations, as well as discussing any assumptions made in the scoring function.

**Questions:**

See above

**Limitations:**

Yes

---

> ### Author Rebuttal · Authors · 2024-08-07
>
> ### Response to Reviewer o6iJ.
>
> We sincerely thank reviewer o6iJ for detailed feedback and we address the reviewer’s concerns as follows.
>
> >Q1: The experiments might not fully capture the diversity and complexity of real-world datasets, which could affect the generalizability of the findings. I recommend conducting additional experiments using a wider variety of datasets to better evaluate the model's performance in real-world scenarios.
>
> Thank you for your valuable suggestions. We have further evaluated our approach on other real-world datasets. We utilized the PubChem dataset from [1]. We sampled 20,000 molecules from it to serve as an auxiliary dataset and employed the other molecules as the pre-training dataset. The specific results are as follows and the performance on the PubChem dataset indicates the broad applicability of our method.
>
> |~|One Step||
> |-|-|-|
> |~|K = 50||
> |~|Prec.|FCD|Prec.|FCD|Prec.|FCD|Prec.|FCD|
> |MLP|0.43|18.26|0.30|**16.75**|0.35|18.32|0.25|**16.52**|
> |Ours|**0.46**|**17.29**|**0.37**|17.42|**0.42**|**16.54**|**0.34**|17.25|
>
> >Q2: The scoring function is based on G, which is a linear combination of the representations of R and M. However, considering that the target model is treated as a black-box model, it is unclear whether there is a linear relationship between the representations of R and M obtained by the black-box model and the representation of G. To clarify this point, I suggest providing a thorough analysis of the relationship between the representations, as well as discussing any assumptions made in the scoring function.
>
> Thank you for your valuable and insightful feedback. Here, we illustrate the rationale behind the scoring function from a theoretical perspective.
>
> Assume that graph $G$ is composed by $G:=G_{1} \cup G_{2}$ and graph pre-trained model is represented by $f$. Let the loss function for the pre-training task be $\mathcal{L}$, which takes graph representation as input. Further, we assume that $\mathcal{L}$ is a bijection and linear mapping. Without loss of generality, we assume that loss function belongs to the category of weighted sum, that is $\mathcal{L}(f(G)) = \alpha_1 \mathcal{L}(f(G_1)) + \alpha_2 \mathcal{L}(f(G_2))$, with $\alpha_1$ and $\alpha_2$ serve as hyper-parameters (This assumption is common among various tasks. For instance, in the most common case of cross-entropy for classification task, $\alpha_1 = |G_1|/|G|$ and $\alpha_2 = |G_2|/|G|$.).
> We can infer that $f(G)=\mathcal{L}^{-1}(\alpha_1 \mathcal{L}(f(G_1)) + \alpha_2 \mathcal{L}(f(G_2)))=\alpha_{1}f(G_{1})+\alpha_{2}f(G_{2})$.
>
> Given the theoretical analysis, we can observe that the relationship between $f(G)$, $f(G_{1})$, and $f(G_{2})$ is akin to a weighted combination, which justifies our design of score function.
>
> Reference:
>
> [1] Y. Wang, J. Wang, et al. "MolCLR: Molecular Contrastive Learning of Representations via Graph Neural Networks." Nat. Mach. Intell., 2022.

---

> ### Author Response · Authors · 2024-08-11
> **Any unanswered questions yet?**
>
> Dear Reviewer o6iJ,
>
> Thanks again for your detailed review and questions. We have provided detailed answers to them in the rebuttal. As we are approaching the end of the discussion phase, we would like to kindly ask if you still have any unanswered questions about our paper?
>
> Best, Authors

---

> > ### Comment · Reviewer_o6iJ · 2024-08-12
> >
> > Thank you for the responses. And I have also read other reviewer's comments. The rebuttal has addressed some of my concerns. I have raised my rating accordingly.

---

### Official Review · Reviewer_JnnP · 2024-07-12

**Soundness:** 3
**Presentation:** 3
**Contribution:** 4
**Rating:** 7
**Confidence:** 4

**Summary:**

This paper investigates the issue of data leakage risks when using pre-trained molecular models in a shared environment and proposes a method to extract training data from such pre-trained models. Specifically, the authors employ a molecular generation method based on templates and a candidate motif bank to attempt to generate potential training data. By designing a scoring function, the model can distinguish substructures that belong to the training data, and using a policy-based reinforcement learning method, the exploration space for generation is narrowed. Finally, the authors evaluated the method on various models trained using ZINC15 dataset.

**Strengths:**

- The issue mentioned in this paper is important and currently overlooked in the community. I appreciate the authors for bringing up this problem and making an initial attempt to address it.

- The proposed method is innovative and satisfactory. Although it does not involve complex model structures, the problem is clearly defined, and the designed modules are targeted at solving the problem. Particularly, analyzing and understanding potential training data from the representation space and then generating based on certain rules make this modeling design interesting to me.

- The experiments selected various pre-training strategies, demonstrating the effectiveness of private data extraction in a cross-model scenario. The conclusions drawn by the authors indicate a significant risk of data leakage.

**Weaknesses:**

While I recognize the research significance and model solution of this work, some design aspects could be improved, specifically:

- Regarding the selection of template structures, the paper ultimately chooses rings as the starting structures for subsequent generation/growth. However, a proportion of molecular data does not contain rings, so this heuristic choice of starting templates might not be the best strategy. The authors could consider constructing a template bank to avoid this issue.

- Are the 20k molecules used in the experiments on line 269 also obtained from ZINC15? Is this a strong iid assumption? I would like to see the effect of data extraction when the auxiliary dataset is replaced with molecules having significantly different distributions, for example, selecting some molecules from the PubChem dataset.

- The meaning of "step" in Table 1 seems unclear. Is it referring to the time-step mentioned earlier? If so, why only explore the case of one and two steps? What characteristics would molecules generated with more steps have? This seems to lack discussion and analysis.

- The author explores few model backbones, so I suggest the authors also conduct experimental evaluation on Graph Transformer or other architectures. And the pertaining data set is also limited, which would be more convincing if another one dataset was added.

**Questions:**

A key metric in the experimental evaluation is Precision. How do you determine whether the generated molecules exist in the dataset? Since molecules lack unique identifiers and it is easy to consider two identical molecules as different mistakenly, I hope the authors can provide clearer details on how the presence of data is judged.

**Limitations:**

Please refer to Weaknesses and Questions.

---

> ### Author Rebuttal · Authors · 2024-08-07
>
> ### Response to Reviewer JnnP.
>
> We greatly appreciate reviewer JnnP for the time and effort you have dedicated to reviewing our paper. We address the reviewer's concerns as follows:
>
> >Q1: Regarding the selection of template structures, the paper ultimately chooses rings as the starting structures for subsequent generation/growth. However, a proportion of molecular data does not contain rings, so this heuristic choice of starting templates might not be the best strategy. The authors could consider constructing a template bank to avoid this issue.
>
> Thanks for your valuable suggestion. We first clarify rationality behind choosing ring structures as starting template is that the ring structures are very common in chemical datasets. In the ZINC15 dataset with 2 million unlabeled molecules, we identified 1,990,890 molecules containing ring structures, constituting the majority of the dataset. Furthermore, the diversity of ring structures offers a wide range of options for template structures (such as tetrahydrofuran and cyclobutane). Therefore, we have implemented a template bank that incorporates the 84 most frequently occurring rings from the auxiliary dataset.
>
> We also appreciate your suggestions to explore template structures that do not contain rings.  We further incorporate five ring-free scaffold templates (i.e., alkanes) into the template bank. The performance are shown as follows. It can be observed that incorporating these ring-free templates has led to slight improvement in performance.
>
> |~|One Step||||Two Step||||
> |-|-|-|-|-|-|-|-|-|
> |~|K = 50||K = 100||K = 100||K = 200||
> |~|Prec.|FCD|Prec.|FCD|Prec.|FCD|Prec.|FCD|
> |Ours|0.50|19.22|0.35|**19.85**|0.31|23.57|0.51|**23.09**|
> |Ours-ring|**0.51**|**19.11**|**0.36**|20.18|**0.37**|**23.27**|**0.52**|23.18|
>
> >Q2: Are the 20k molecules used in the experiments on line 269 also obtained from ZINC15? Is this a strong iid assumption? I would like to see the effect of data extraction when the auxiliary dataset is replaced with molecules having significantly different distributions, for example, selecting some molecules from the PubChem dataset.
>
> We first clarify that 20k molecules employed for the auxiliary dataset were also sourced from ZINC15. In Appendix A.3, we have illustrated the overlap between the pre-training dataset and the auxiliary dataset. We found only 103 identical molecules, indicating a relatively small overlap ratio, which suggests it does not strongly adhere to the IID assumption.
>
> Besides, following the reviewer's insightful suggestion, we sampled 20,000 molecules from PubChem [1] as auxiliary dataset to ensure a difference from the ZINC pre-training dataset. The results based on GraphCL pre-trained model are shown as follows. We observed that under these scenarios, performance of our method have a slight decline. However, it still demonstrates comparable efficacy and showcases the robustness, and this observation aligns with what we presented in Appendix A.4.
>
> |~|One Step||||Two Step||||
> |-|-|-|-|-|-|-|-|-|
> |~|K = 50||K = 100||K = 100||K = 200||
> |~|Prec.|FCD|Prec.|FCD|Prec.|FCD|Prec.|FCD|
> |Ours|0.50|19.22|**0.35**|19.85|**0.31**|23.57|**0.51**|**23.09**|
> |Ours-PubChem|**0.52**|**18.52**|0.33|**17.90**|0.26|**20.57**|0.31|24.41|
>
> >Q3: The meaning of "step" in Table 1 seems unclear. Is it referring to the time-step mentioned earlier? If so, why only explore the case of one and two steps? What characteristics would molecules generated with more steps have? This seems to lack discussion and analysis.
>
> Apologize for not being clear and appreciate for providing an opportunity to clarify. The "step" mentioned here refers to the time-step as understood by the reviewer. The reason for selecting only one and two steps is that other baseline methods have difficulty managing molecular structures beyond two steps, due to the increase in complexity. Consequently, we confined our comparisons to just one and two steps.
>
> However, our approach based on reinforcement learning can be extended to multiple steps. Here, we extend the time-step, and results are as follows (we only take "Random" as baseline considering the runtime of other baselines). It can be observed that as the time-step increases, performance may decline due to the increased difficulty of extraction caused by the complexity of the molecules. Nevertheless, our model still outperforms the baseline model in precision, indicating the effectiveness of our extraction method.
>
> |~|One Step||Two Step||Three Step||Four Step||
> |-|-|-|-|-|-|-|-|-|
> |~|$K=50$|$K=100$|$K=100$|$K = 200$|$K=100$|$K = 200$|$K=100$|$K = 200$|
> |Random|0.05|0.09|0.09|0.07|0.02|0.00|0.00|0.00|
> |Ours|**0.50**|**0.35**|**0.31**|**0.52**|**0.27**|**0.25**|**0.17**|**0.18**|
>
> Table:Precision of molecular extraction results among different time step

---

> ### Author Response · Authors · 2024-08-07
> **Additional Rebuttal**
>
> >Q4: The author explores few model backbones, so I suggest the authors also conduct experimental evaluation on Graph Transformer or other architectures. And the pertaining data set is also limited, which would be more convincing if another one dataset was added.
>
> Sorry for not explaining clearly. In our experiments, we have employed various encoder architectures for molecular pre-trained models. The Grover graph pre-trained model included in Table 1 is based on the GTransformer [2], a Graph Transformer. The performance demonstrates the effectiveness of our model across various pre-trained model architectures.
>
> As for the limitation of pre-training dataset, we further introduced a new dataset. We employed the PubChem dataset used in [1]. We sampled 20k molecules from the PubChem dataset to serve as the auxiliary dataset and employed the other molecules as the pre-training dataset. The specific results are as follows and the performance on the PubChem dataset indicates the broad applicability of our method.
>
> |~|One Step||
> |-|-|-|
> |~|K = 50||
> |~|Prec.|FCD|Prec.|FCD|Prec.|FCD|Prec.|FCD|
> |MLP|0.43|18.26|0.30|**16.75**|0.35|18.32|0.25|**16.52**|
> |Ours|**0.46**|**17.29**|**0.37**|17.42|**0.42**|**16.54**|**0.34**|17.25|
>
> >Q5: A key metric in the experimental evaluation is Precision. How do you determine whether the generated molecules exist in the dataset? Since molecules lack unique identifiers and it is easy to consider two identical molecules as different mistakenly, I hope the authors can provide clearer details on how the presence of data is judged.
>
> Sorry for not explaining it clearly. We utilized the RDKit library to determine molecular identity [3]. Specifically, it assesses whether there exists an atomic mapping such that each atom in the query molecule can be paired with a corresponding atom in the target molecule, with the same connectivity as in the query. Detailed explanation will be included in the revised version.
>
> Reference:
>
> [1] Y. Wang, J. Wang, et al. "MolCLR: Molecular Contrastive Learning of Representations via Graph Neural Networks." Nat. Mach. Intell., 2022.
>
> [2] Rong, Yu, et al. "Self-supervised graph transformer on large-scale molecular data." NeurIPS, 2020.
>
> [3] Landrum, Greg. "RDKit: A software suite for cheminformatics, computational chemistry, and predictive modeling." 2013.

---

> ### Author Response · Authors · 2024-08-11
> **Any unanswered questions yet?**
>
> Dear Reviewer JnnP,
>
> Thanks again for your detailed review and questions. We have provided detailed answers to them in the rebuttal. As we are approaching the end of the discussion phase, we would like to kindly ask if you still have any unanswered questions about our paper?
>
> Best, Authors

---

> > ### Comment · Reviewer_JnnP · 2024-08-11
> >
> > Thanks for the response. I've read through other reviewers' feedback and responses as well. I believe this work holds practical value, so I have provided continued support. Although, as other reviewers have mentioned, the method itself may not be particularly fancy, I think it offers a new perspective and a feasible solution for safely applying molecular pretraining models in collaborative scenarios.
> >
> > Therefore, I will increase my rating to `7`.
> >
> > Best regards,
> >
> > Reviewer JnnP

---

### Decision · Program_Chairs · 2024-09-25

**Decision:**

Accept (poster)

**Comment:**

All reviewers except one (qKJK) argued for accepting the paper. For qKJK their main conerns were on (a) how realistic the problem setup is, (b) the score function lacking theoretical or chemical justification, (c) missing regression tasks. The authors respond convincingly to each point. For (a) they point out, as the reviewer does, that sharing a classification/regression model poses a higher privacy risk because of the potential leakage of confidential label data. For this reason there are at least two papers which have proposed sharing graph representation learning models instead. Two other papers have also highlighted the necessity for a generic pre-trained representation learning model. As far as I am aware this paper is the first to point out the issue of extracting training data from these models and to give an attack to extract such data. The authors give a convincing real-world example of a collaboration between 10 pharmaceutical companies who jointly trained a model for structure-activity. They cite a paper that describes how the companies had privacy concerns regarding their molecule structures, treating them as confidential. They cite two other papers on how molecular structures are considered sensitive IP in various settings.. For (b) they provide a better intuition of the score function, and for (c) they provide new results on a regression tasks. In response, Reviewer qKJK still argued against (a), saying that the motivation is not realistic. They argue that because there are large public databases of molecular structure then this shouldn’t be an issue. This completely ignores the author’s response that there are many scenarios where molecular structure privacy is an issue. Therefore, I find this point resolved. As all concerns are resolved, I vote to accept. Authors: you’ve already made improvements to respond to reviewer changes, if you could double check their comments for any recommendation you may have missed on accident that would be great! The paper will make a nice contribution to the conference!